# "It's not that I don't trust vaccines, I just don't think I need them": Perspectives on COVID-19 vaccination

**Catherine Pelletier**[1], **Dominique Gagnon**[2], **Eve Dubé**[1,2,3]*

1 Centre de recherche du CHU de Québec–Université Laval, Québec, Québec, Canada, 2 Institut national de santé publique du Québec, Québec, Québec, Canada, 3 Département d'anthropologie, Université Laval, Québec, Québec, Canada

* eve.dube@ant.ulaval.ca

**Data Availability Statement:** All relevant data that can be publicly displayed are available within the paper and its Supporting Information files. According to the information and consent form

## Abstract

In Quebec (Canada), the roll-out of the vaccination started slowly in December 2020 due to limited vaccine supply. While the first and second doses were well-accepted among adults and vaccine uptake was above 90%, in late 2021 and 2022, vaccine acceptance decreased for children vaccination and receipt of a 3rd or a 4th dose. In the autumn of 2022, four focus groups were conducted with vaccine-hesitant parents of children aged 0–4 and adults who expressed little intention to receive a booster dose. The objective of this study was to gather participants' perspectives on vaccination in general, on the COVID-19 vaccination campaign and the information available, and to gain insights into the underlying reasons for their low intention of either having their child(ren) vaccinated, or receiving an additional dose of vaccine. A total of 35 participants took part in the focus groups. While participants expressed a certain level of trust and confidence in public health and government authorities regarding pandemic management and the vaccination campaign, they were also concerned that transparent information was lacking to support an informed decision on booster doses and children's vaccination. Many participants felt adequately protected against the infection during the focus groups, citing a lack of perceived benefits as the primary reason for refusing a booster dose. Parents who refused to administer the COVID-19 vaccine to their young children felt that the vaccine was not useful for children and were concerned about potential side effects. The majority reported that their opinions regarding other recommended vaccines had not changed since the beginning of the pandemic. While these results are reassuring, our findings highlight the importance of transparency in public health communications about vaccines to increase confidence and to develop strategies to address vaccine fatigue and complacency toward COVID-19 vaccines.

## Introduction

In Canada, the first COVID-19 vaccines were available in late 2020, making the country one of the first to offer COVID-19 vaccination to its population. At the beginning of 2021, different studies showed a strong intention to receive a COVID-19 vaccine in the population, with less

signed by participants, and in accordance with the ethical principles of the CHU de Québec-Université Laval Ethics Committee, only the principal investigator and his team have access to project materials, including information and consent forms, recordings and videos, and data in digital format. Therefore, we are not permitted to share any additional data beyond the anonymized quotations presented in the submitted manuscript. If a researcher wishes to access data, he/she can contact the CHU de Québec-Université Laval Research Ethics Committee via email (gurecherche@chudequebec.ca).

**Funding:** This research was funded by the Quebec Ministry of Health (https://www.msss.gouv.qc.ca/). ED received the grant. The funder had no role in study design, data collection and analysis, decision to publish, or preparation of the manuscript.

**Competing interests:** The authors have declared that no competing interests exist.

than 15% of Canadian adults not intending to receive a COVID-19 vaccine when available [1, 2]. In a time when vaccination was seen as key to restoring normalcy, both government and public health authorities undertook an immense effort to implement an unprecedentedly large-scale vaccination campaign.

In Quebec, the COVID-19 vaccination campaign began in late December 2020. Quebec is the second most populous province in Canada and is predominantly French-speaking. Canada is a federation and healthcare is under provincial and territorial jurisdiction. While national authorities were tasked with setting public health guidelines and overseeing vaccine supply and distribution during the pandemic, each province issued its own vaccination recommendations and managed its vaccination campaign. In Quebec, as a limited number of vaccine doses were available at the time, vaccination was initially offered to individuals at higher risk of contracting COVID-19 or developing complications (i.e., residents of long-term care centers, healthcare workers, people living in isolated communities) [3]. As more and more vaccine doses were available, the campaign was extended to the general adult population in March 2021 on a progressive basis, by age group. Following Health Canada's approval of a COVID-19 vaccine for adolescents aged 12 to 17, vaccination was then offered to this group as of May 2021. In early summer 2021, Quebecers showed a positive response to COVID-19 vaccination efforts, with over 75% of individuals aged 12 and over receiving at least one vaccine dose [4, 5]. As new variants started to emerge and circulate [6], the province implemented various measures during the summer of 2021, including incentives such as vaccine lottery and vaccine passport to increase the uptake of the second dose and encourage unvaccinated individuals to get vaccinated [7, 8]. In November 2021, vaccination was extended to children aged between 5 and 11 years, concurrently with the administration of a 3rd dose (hereafter named "booster dose"), which was initially offered to priority groups. In the aftermath of the Omicron wave, a second booster dose was also recommended at the end of March 2022 to all adults, shortly after the withdrawal of the use of the vaccine passport in the province.

In the summer of 2022, the COVID-19 vaccination was made available for children aged 6 months to 4 years based on a discretionary recommendation to parents [9]. Acceptability issues in this age group were anticipated. Indeed, COVID-19 vaccination of children aged 5 to 11 years had already received a mixed reception from parents in Quebec, with, as of November 2022, 56% of children in this age group having received two doses of COVID-19 vaccine [10]. One year after the vaccine became available for this age group, more than a quarter of parents still expressed no intention of having their children vaccinated against COVID-19 [10]. Reasons frequently reported by parents included low-risk perception of COVID-19 and concerns about possible long-term vaccine side effects in children [11, 12]. Similar to other international studies, which have shown lower intention rates among parents of very young children [11], it was unsurprising to find that nearly two-thirds of Quebec parents had no intention of vaccinating their children aged 6 months to 4 years against COVID-19 in November 2022 [10].

On August 16, 2022, a fall vaccination campaign against COVID-19 was also launched, to boost Quebecers' immunity to COVID-19 before the return to school and the colder season [13]. At this point, signs of vaccine fatigue were observed in those who already had received two, three, or four doses of vaccine since the beginning of the campaign. For example, in November 2022, 77% of Quebecers had completed their primary vaccination (2 doses), and only 28.6% had received an additional dose following the launch of the 2022 fall vaccination campaign [4, 5]. Surveys conducted during the same period also showed that less than half of Quebecers intended to receive an additional dose during the 2022 fall vaccination campaign [10].

In this context, this qualitative study aimed to better understand the reasons behind the low acceptance of COVID-19 vaccination for children aged 6 months to 4 years, as well as the reasons behind the low intention for an additional dose of the vaccine.

## Material and methods

### Study design, recruitment, and sampling

We used a qualitative focus group study design. Two categories of participants were recruited: 1) parents with children under 5 years of age, and 2) Quebec adults. Two focus groups were planned for each group of participants.

Participants were recruited from a pool of respondents to web surveys who indicated a willingness to participate in qualitative studies after completing the questionnaire. Recruitment and administration of the surveys were carried out by a specialized research firm [14] and details about the survey's methodology are available elsewhere [15].

Diverse groups of participants were invited to participate in the focus group discussion among survey respondents who had consented to be contacted again. The specialized research firm was responsible for contacting, selecting, and recruiting participants who met the inclusion criteria. Thus, to be eligible to participate in the focus groups on the acceptability of COVID-19 vaccination for children, participants needed to meet the following criteria at the time of recruitment: 1) they had to be parents with at least one child under the age of 5, 2) their child(ren) must not have received the COVID-19 vaccination, and 3) they had to either express no intention, be hesitant, or be unsure about vaccinating their child(ren). For the focus groups on the acceptability of a booster dose of a COVID-19 vaccine, participants had to meet these criteria at the time of recruitment: 1) they needed to have received either two or three doses of a COVID-19 vaccine, and 2) they had to either express no intention, be hesitant, or be unsure about receiving an additional dose in the fall of 2022.

### Data collection

Focus groups (n = 4) took place virtually between November 8, 2022, and November 15, 2022. Facilitation was handled by a professional with expertise in moderating focus groups, hailing from the same specialized research firm that managed the recruitment process. A semi-structured interview guide covering topics related to trust in government and public health authorities during the pandemic, vaccination (COVID-19 and routine), and information available on COVID-19 vaccination was used. Participants were invited to share whether they intended or not to vaccinate their child(ren) aged 5 and under or to receive a booster dose themselves. Sessions lasted approximately 90 minutes and were audio-recorded. A financial compensation of $100 was provided to each participant based on the research firm's usual practices.

### Data analysis

The auto recordings were transcribed. A thematic content analysis was conducted using NVivo 12 software to identify and interpret themes within the dataset [16]. Inductive and deductive approaches were used to generate codes. While certain themes were pre-identified based on the discussion guide, additional themes emerged during the analysis to capture participants' perspectives. Open coding was first performed by a member of the research team (CP) and the preliminary codes were reviewed and validated by another research team member (DG). Analysis was first conducted separately for the two groups of participants and then merged to identify similarities and differences in participants' opinions.

### Ethical considerations

The CHU de Québec-Université Laval ethics committee has approved this study (2023–6599). Prior to the discussions, written consent was obtained from all participants.

**Table 1. Sociodemographic characteristics of participants.**

| | Parents | | Booster dose | |
|---|---|---|---|---|
| | Group 1 | Group 2 | Group 1 | Group 2 |
| **N** | 9 | 8 | 10 | 8 |
| **Age** | | | | |
| 18 to 34 years | 5 | 2 | 5 | 6 |
| 35 to 54 years | 4 | 6 | 4 | 2 |
| 55 years and older | 0 | 0 | 1 | 0 |
| **Gender** | | | | |
| Female | 6 | 4 | 4 | 5 |
| Male | 3 | 4 | 6 | 3 |
| **COVID-19 vaccination status[a]** | | | | |
| Vaccinated with 2 doses | 5 | 0 | 3 | 3 |
| Vaccinated with 3 doses | 4 | 6 | 6 | 5 |
| Vaccinated with 4 doses | 0 | 2 | 1 | 0 |
| **Intention to vaccinate child(ren) under 5 years of age against COVID-19[a]** | | | | |
| Intention | 0 | 4 | - | - |
| Has no intention | 8 | 2 | - | - |
| Hesitant | 1 | 2 | - | - |
| **Intention to receive a COVID-19 booster dose[a]** | | | | |
| Intention | - | - | 0 | 0 |
| Has no intention | - | - | 6 | 8 |
| Hesitant | - | - | 2 | 0 |
| Intention unspecified | - | - | 2 | 0 |

[a] As reported during the focus groups.

## Results

Overall, 35 participants joined one of the focus groups, with 17 participants in focus groups specific to COVID-19 pediatric vaccines and 18 participants, to COVID-19 booster vaccines (Table 1). Most participants were aged between 25 and 44 years, and there was a slightly higher representation of women. Participants' intentions regarding COVID-19 vaccination varied, but a low intention was expressed for both COVID-19 pediatric vaccines and booster doses. All participants reported having received either two or three doses of COVID-19 vaccine. Additionally, almost all participants reported having contracted COVID-19 at least once since the start of the pandemic (information not presented in Table 1).

The main topics covered in the focus groups were aligned with the themes of the semi-structured interview guide. The findings are presented in an aggregated manner and are divided into four sections: opinion (including intention) towards COVID-19 vaccines, perceptions of routine vaccines since the start of the pandemic, trust in authorities and perceptions of vaccine campaign and related measures, and attitudes and preferences in terms of information about COVID-19 vaccines. We paid specific attention to similarities and differences in discourses about COVID-19 booster doses and the vaccination of children against COVID-19.

### Participants' opinions towards COVID-19 vaccines

While some participants had positive opinions about COVID-19 vaccines' safety and efficacy, evoking their general confidence in science and vaccination, others had significant concerns

regarding the novelty of the vaccine and the lack of experience with long-term side effects. These concerns were even greater regarding the vaccination of children under 5 years old, with many concerned about possible future vaccine mandates for this age group. COVID-19 vaccine safety was also an important source of concern, especially among participants who had become pregnant after the launch of the COVID-19 vaccination campaign. These participants expressed the difficulty they had in deciding to receive the vaccine, and some reported having conflicting information about the safety of COVID-19 vaccination during pregnancy. Other concerns related to vaccination centered around the challenge of balancing between the risks of long COVID and those associated with COVID-19 vaccination.

**Participants' intentions to vaccinate their child(ren) aged less than 5 years old against COVID-19.**  Most of the parents had no intention of vaccinating their child(ren) under 5 years old against COVID-19. The main reason was the perceived lack of benefits. Many participants saw no advantage in vaccinating young children that are not severely affected by the disease, and with vaccines that do not prevent transmission of the virus. Other parents pointed out that children in this age group are frequently exposed to different viruses (in daycare, in contact with other children in the family, etc.) which enables them to develop their immune systems naturally. Furthermore, parents whose children have had COVID-19 reported mild symptoms, reinforcing their lack of perceived benefits of the vaccine.

Parents who reported an intention to have their child(ren) vaccinated against COVID-19 were motivated by the desire to offer additional protection to their child, as well as to those around them. During the discussions, a few parents were still ambivalent about whether to vaccinate their child(ren) or not. Their hesitation was mainly related to their perception of a lack of information regarding the effects (known and unknown) of the vaccine, which they believed was developed too rapidly. In the event of a more COVID-19 serious situation, such as an increase in the number of cases, some hesitant parents stated that they would reconsider their decision to eventually accept the vaccine.

**Participants' intentions to receive a COVID-19 booster dose.**  The majority of participants had no intention to receive an additional dose during the fall vaccination campaign. Most participants had also already contracted COVID-19 at least once and had experienced only mild symptoms, which contributed to their perceptions that an additional dose of vaccine was unnecessary. Doubts about the efficacy of the vaccines were also expressed among participants, particularly by those who contracted the disease after vaccination. Finally, few participants mentioned having suffered from significant side effects following the first doses of vaccine justifying their refusal to receive a booster dose. Different reasons were reported by the few participants who already had received a first booster dose, such as the perception that a third dose would be needed for the vaccine passport, or because the third dose was strongly recommended for them (e.g., in preparation for surgery, participants with underlying medical conditions, pregnant participants).

Participants were also asked about different incentives and the impact these might have on their decision to accept or not a booster dose in the future. Although many of the participants explicitly expressed their lack of intention to accept an additional dose of the COVID-19 vaccine, the emergence of a new wave of COVID-19, updated vaccination requirements, and the necessity of a booster dose for travel were identified as potential reasons that could sway their intention. The recent availability of a bivalent vaccine was not perceived as a sufficient incentive to convince participants to seek a booster dose. Many mentioned that new variants were constantly appearing which would not have been included in this new formulation.

Examples of quotes reflecting participants' views on COVID-19 vaccines*

- I have no intention of vaccinating my children. It's completely irrational. But they've had all the other vaccines. I'm vaccinated against COVID-19, but I don't know, I'm not ready for that. If there are no benefits, even if there are no risks. . . As long as there are no significant advantages, I don't see the point of going there, in the current situation of course, about COVID-19. If things evolve, if things change, if there's a need, I could reconsider. But for the moment, I don't see the point. (Participant 4, COVID-19 vaccine for children, Focus group 2)

- Yes, I intend to have him vaccinated. For him, a little, but mostly for the people around him who might contract it. It's more for the people around him than for him directly. Because I think that's the big advantage at his age. (Participant 3, COVID-19 vaccine for children, Focus group 2)

- As these are new vaccine methods that were developed in a hurry. I agree with [other participant], at the time, it was worrying both for me and for the future, for the children. That's why I chose not to vaccinate them. Because I don't know what it will do in 10- or 20 years, unlike the vaccines we're used to. That was my biggest concern. (Participant 3, COVID-19 booster dose, Focus group 1)

- I've had my 2 doses, but as far as booster doses go, I don't see the point. It's not that I don't trust vaccines, but I don't think I need them. (Participant 7, COVID-19 booster dose, Focus group 2)*All quotes have been translated from French

## Participants' perceptions of routine vaccines since the start of the pandemic

Despite concerns about COVID-19 vaccines and low intentions, opinions of other routine vaccines remained the same as before the pandemic for all participants interviewed, including those involved in COVID-19 booster dose discussions. Their confidence in routine vaccines stemmed from their perceptions of robust scientific evidence demonstrating the long-term efficacy and safety of such vaccines which was not the case for COVID-19. Despite this high level of confidence, some participants still had concerns about specific vaccines administered during childhood or adulthood (e.g., MMR vaccine because of its supposed link with autism in children, varicella vaccine because of its more recent availability, influenza vaccine because of its uneven efficacy).

During the discussions, all parents from all four focus groups spontaneously affirmed that their child(ren) had received all the recommended vaccines up to now and that their vaccination records were up to date. Parents who had their child(ren) vaccinated for routine vaccinations amid the COVID-19 pandemic did not raise any significant system-related barriers, as they perceived that the services provided remained unchanged. However, some parents mentioned additional prerequisites when scheduling appointments due to health protocols (e.g., absence of COVID-19 symptoms, restrictions on the number of individuals admitted to the clinic).

At a more general level, participants also expressed concerns about the potential resurgence of vaccine-preventable diseases due to the rise of anti-vaccine rhetoric, amplified by the polarization caused by the COVID-19 vaccine. A participant mentioned having relatives who used to vaccinate their children against vaccine-preventable diseases who have become anti-vaccine since the COVID-19 vaccination campaign, now refusing all vaccines.

### Examples of quotes reflecting participants' views on routine vaccines*

- It's because it's a new vaccine. Other vaccines have been tried and tested. There aren't many side effects. The research is there. And it's been years. It's probably the same vaccines I had when I was a kid, 30 years ago. They've proved their worth. The vaccine for COVID is brand new. If they came out with a new vaccine for another disease X, I'd be just as skeptical. (Participant 9, COVID-19 vaccine for children, Focus group 1)

- Both my children are up to date with their vaccinations. My youngest, in the beginning, during the big lockdown, the first one, that's when her vaccines were most due. Then I didn't have any problems. I think I had the least trouble getting appointments for routine vaccinations because for the doctor, it was difficult. Everything was difficult. But there was never any problem with the vaccination, it went well. (Participant 1, COVID-19 vaccine for children, Focus group 1)

- What I mean is, I think COVID-19 vaccine talk has polarized people and made them more anti-vax. And because of this, we might start to see a resurgence of old diseases, because there would be more unvaccinated children against vaccine-preventable diseases. (Participant 8, COVID-19 vaccine for children, Focus group 2)

- But I don't take the flu shot because I'm a bit worried about it. [. . .] Of course, because of my health condition. . . and it's clear that getting an injection. . . It's like injecting an entity into your body, and I think you have to be well-informed. (Participant 5, COVID-19 booster dose, Focus group 1)*All quotes have been translated from French

## Participants' trust in authorities and perceptions of vaccine campaign and related measures

The issue of trust in public health authorities, the government, and vaccination was addressed during the discussions, generating varied opinions among participants across the different groups. Participants were first asked about their perceptions of the management of the COVID-19 pandemic. Most of those who expressed trust in the institutions highlighted the unprecedented nature of the COVID-19 pandemic and the authorities' ability to respond rapidly in such a context, although some participants felt that the government had sometimes seemed overwhelmed by the situation. Moreover, amidst the uncertainties surrounding the progression of the pandemic and the imperative to mitigate health-related consequences, the necessity for trust and adherence to population-level decisions made by stakeholders appeared evident to some participants. However, other participants emphasized that the level of trust held no significance in the implementation of certain measures requiring compliance or adherence. For these participants, trust played a minimal role in determining the level of adherence, as they were following the rules implemented as part of the state of emergency (e.g., curfew, travel bans), regardless of their opinion of the authorities. Finally, some participants reported having little trust in the authorities, particularly due to the perceived lack of transparency and coherence in the decision-making process. Others expressed a decrease in their level of trust over the pandemic, particularly regarding the perceived politicization of certain measures aimed at containing the virus (e.g., lockdowns, curfew).

Regarding COVID-19 vaccination, many participants expressed confidence in vaccines, although some mentioned that having trust did not exclude "critical thinking". Some participants mentioned that the short duration of protection conferred by COVID-19 vaccination had negatively impacted their level of trust in this measure. While several participants reported positive views about the roll-out of the vaccination campaign in the province (e.g., user-friendliness of the online appointment platform, and efficiency at vaccination sites), many felt compelled to get vaccinated, especially following the implementation of the vaccine passport. Others thought that the authorities had played on peoples' feelings to enforce vaccination. Some also expressed a perceived social pressure (e.g., by relatives or colleagues) to adhere to vaccination or at least to get the first two doses of the vaccine. Although some participants reported having initially supported the implementation of incentives and disincentive measures to encourage people to get vaccinated, they later changed their opinion as it became evident that the vaccination did not have the desired effect on the progression of the pandemic. Other participants mentioned the concept of free choice regarding vaccination decisions and negatively perceived the use of coercive measures related to COVID-19 vaccination to increase adherence (e.g., mandates).

### Examples of quotes reflecting participants' views on the management of the COVID-19 pandemic and vaccination campaign*

- At the beginning, I think it went really well. They were well organized, given that there wasn't much stock to vaccinate everyone. But right now, I think the movement is running out of steam. People are less inclined to get vaccinated. I've had 3 doses. [. . .] And now I've got COVID. I've caught it. I'm doing fine. I'm much better today. All that to say that maybe I'm a bit isolated, but I have the impression that people around me are running out of steam with this vaccination as if they were wondering: "Is it still as effective as it was at the beginning?" (Participant 1, COVID-19 vaccine for children, Focus group 2)

- What I don't like about the vaccination campaign, at the beginning they proposed to people, and they encouraged people to do it. But like a lot of people, when they started forcing people with threats of "you can't go to such and such a place, you can't work", even in hospitals, they wanted to remove people who weren't vaccinated. I kind of understand the principle. For me, it was less of an incentive to get vaccinated, and I'm not against vaccines in life. I've got all my vaccinations up to date. But to make me feel obliged to do it, I didn't like the feeling it gave me. (Participant 1, COVID-19 vaccine for children, Focus group 1)

- I trust the public health authorities. It's like an unusual situation. It's a new situation. They give information as best they can, when they have it, with the information they have. Scientific data evolves. Information evolves. So they do the best they can to the best of their knowledge at the time of the data. Then there's also the political side of things, which is a little. . . disturbing. But I tell myself that there are people behind it, people with a professional conscience. At some point, decisions must be made. (Participant 2, COVID-19 booster dose, Focus group 2)

- Trust waned a bit. At the beginning, they said that with the vaccine, everything would be fine, we'd go back to the way things were before. Then you get the vaccine, and they

> say, "Oh no, you need a 2nd dose and stay at home". I say to myself: "If they weren't sure at the outset whether it was going to work or not, they were better off just telling us: 'We'll see how it goes as we go along', instead of promising us straight away that everything was going to be sorted out. In the end, it wasn't. So I'd say for me it's been downhill from the start. (Participant 8, COVID-19 booster dose, Focus group 2)*All quotes have been translated from French

## Participants' attitudes and preferences toward available information about COVID-19 vaccines

Participants were asked about their information-seeking habits regarding COVID-19 vaccination. In general, traditional media and official sources (e.g., public health, government) were the most common sources of information mentioned by the participants. A few also mentioned having done online research, while being aware of the diversity in quality and reliability of information available on the Internet. Many parents also identified their pediatrician or family doctor as their preferred source of information. However, some noted that healthcare professionals did not necessarily have additional information beyond what was already available in the mainstream media. They drew a comparison with the availability of information on routine vaccines, which can be obtained from various sources apart from the media or government websites (e.g., leaflets in local health clinics—CLSCs). Other participants expressed little interest in information on COVID-19 vaccines. Some parents explicitly mentioned the fatigue that had set in since the start of the vaccination campaign, while participants in the dose booster focus groups said they had not sought out information on COVID-19 vaccination, relying instead on public health recommendations.

Despite the abundance of information disseminated during the vaccination campaign, most participants in both the 6-months-4-years and booster dose groups felt that there was still a lack of truly useful information that would support informed decision-making (e.g., lack of clarity regarding the benefits of vaccinating under 5 years old and the effectiveness of one or more booster doses). A few participants were more critical of the media discourse on COVID-19 vaccination, noting the absence of debate on the subject, especially at the onset of the vaccination campaign.

Finally, participants named their unmet information needs on COVID-19 vaccination. Parents of children under 5 years old reiterated the importance of obtaining information from trusted sources, such as their family doctor or pediatrician. They believed that receiving information from these healthcare professionals would increase confidence in their decision-making. Many participants also agreed that a simple recommendation from public health authorities might not be sufficient and that the usefulness and benefits of vaccination for this age group should be established (e.g., presenting the benefits of vaccinating younger children from a population point of view or in terms of protecting the most vulnerable was not seen as compelling enough). In the booster dose focus groups, some participants requested evidence regarding the efficacy of a booster dose. Others felt that, given the current stage of the pandemic and the vaccination campaign, they did not need any additional information and that the decision mainly relied on personal motivation and choice. This opinion was also shared by some parents.

Examples of quotes reflecting participants' views on COVID-19 vaccines information*

- When vaccines first came out, it was a subject we talked about a lot. Everyone asked, "Have you been vaccinated? "Which vaccine have you had? "How many doses? It was a hot topic. I was pregnant, I wasn't vaccinated, so everyone was saying, "Well, what are you doing not being vaccinated? "It's dangerous for the baby, what are you thinking? In any case, we talked about it a lot. Since this summer, even before that, I don't hear anyone talking about vaccination anymore. (Participant 5, COVID-19 vaccine for children, Focus group 1)

- For routine vaccination, the doctor can provide information. There are leaflets. You can even go to the CLSC, and they'll give you sheets with all the information. There's a lot of documentation. It's easy to find information in health manuals or magazines. Whereas with COVID vaccination, the information you find is either on the government website, what they say in the press, or information like more conspiracy-oriented stuff on the internet, if you do some research. It's more difficult to get informed, truly scientific information about this. (Participant 1, COVID-19 vaccine for children, Focus group 1)

- I don't think they explain enough. Maybe if they gave us some comparisons, like those who didn't get their last dose, if there were more who were hospitalized or not? And those who had the last dose, were they hospitalized less? I don't know. They say to go ahead, that it's better for everyone, but we have nothing to compare it to. (Participant 8, COVID-19 booster dose, Focus group 1)

- For my part, I don't need any more information to do it. We're going to do it. The decision's been made, it's not a problem. But if I must give my opinion on it, I'd say. . . I understand the advantage it has for the community, for the people he'll be around, but if I could have more information on the benefits for him, for my boy, that's what I'd need as additional information. (Participant 3, COVID-19 vaccine for children, Focus group 2)*All quotes have been translated from French

## Discussion & implications

In our study, we found that both the intention to vaccinate children against COVID-19 and the willingness to receive a booster dose were low, with common underlying issues.

One prominent issue that emerged from our analysis is confidence, an important determinant of vaccine hesitancy, as defined in the 5C model [17]. Within this framework, confidence refers to trust in the effectiveness and safety of vaccines, the authorities responsible for making vaccine policies, and the public health science in general [18]. While the majority of our participants initially expressed trust in public health authorities and the government's management of the pandemic and the vaccination campaign, this trust eroded as the vaccination campaign progressed in the province. The main reasons reported were a lack of transparency from authorities regarding the rationale behind the implementation of coercive measures (e.g., curfew, travel bans, vaccine passport). Several studies have identified distrust in authorities and the healthcare system as an important determinant of COVID-19 vaccine hesitancy [19]. These studies have also highlighted a progressive erosion of trust in science and institutions

during the pandemic, driven by a growing perception of inconsistencies in the public health recommendations and measures implemented over time [19–21]. During the pandemic, several experts pointed out that certain coercive measures, such as the vaccine passport or compulsory vaccination, can harm confidence and vaccine intention as they can provoke social and political resistance [22]. In our study, many participants confessed that the vaccine passport had a negative impact on their vaccine confidence and trust in authorities. Our results also revealed that the decline in the protection conferred by COVID-19 vaccines against new variants and over time had a negative influence on participants' confidence in these vaccines, potentially leading to a decrease in willingness to receive a booster dose. Perceived efficacy of the vaccine remains one of the most significant reasons for people to get vaccinated, either for primary COVID-19 vaccine series or booster dose(s) [19, 23, 24]. Participants' perceived risk of COVID-19 also decreased during the pandemic, particularly if they had already received vaccine doses or contracted the virus. Other studies have shown that prior COVID-19 infection could negatively influence COVID-19 vaccine willingness [19, 25]. Low-risk perception of the threat of COVID-19 and low perceived efficacy of the vaccines are known determinants of COVID-19 vaccine hesitancy [24], including for the receipt of booster doses [19, 23, 26]. These factors are also associated with low COVID-19 vaccine acceptance among parents in this study, as well as in others [11, 27–29].

While participants' confidence in COVID-19 vaccine safety and efficacy declined over time, this did not seem to affect their views on other routine vaccines. Most participants remained confident about the importance of routine vaccines, which is reassuring given the lower vaccination rates among children during the early months of the pandemic [30–32]. Fortunately, some data suggest a return to pre-pandemic levels for most vaccines while effective catch-up initiatives have been put in place [30–32]. Nevertheless, public health experts are still concerned about the impact of the pandemic and COVID-19 vaccination on the perception of routine vaccines. A recent UNICEF report highlights a decrease in confidence in routine vaccines across various countries since the pandemic's onset [33]. As vaccine confidence proves to be volatile and subject to fluctuation over time [34], it becomes crucial to adopt a proactive approach and vigilantly monitor the evolution of vaccine hesitancy in the next years [35].

At the time of the study, all public health measures had been lifted in Quebec and information about the pandemic or the vaccine were rarely making the headlines. This context may explain the lack of information that some participants expressed about booster doses or vaccinating their child(ren) against COVID-19. Parents cited limited clear information, especially regarding the benefits of vaccinating children against COVID-19. This lack of access to relevant information was frequently reported in other studies as a reason for vaccine hesitancy among parents [36, 37]. Official sources and healthcare professionals remain essential to enhance vaccine acceptance and uptake in this population, underscoring the need for improved communication and information accessibility.

## Public health implications

While our findings relate to COVID-19 vaccination, we identified potential lessons for future vaccination program implementations (i.e., against respiratory syncytial virus or group B streptococcus). Even if tremendous efforts were made to communicate COVID-19 vaccine recommendations, many participants indicated that the information provided did not fully meet their needs. Efforts to communicate relevant information about vaccines must be more targeted and tailored to address the "information overload" and conflicting messages from different sources [38]. Of note, participants who were pregnant during the campaign were

concerned about vaccines' safety and efficacy in pregnancy. As new vaccines are expected to be offered during pregnancy, ensuring readily available and trusted information, along with equipping maternity care providers to discuss and recommend vaccines, is crucial.

This study has several limitations. Although the focus group method allows us to interview different individuals with varying opinions, it is not feasible to cover all the factors influencing vaccination due to the limited number of questions. Furthermore, the small sample size and the recruitment criteria (i.e., participants with either no firm intention to vaccinate their child (ren) or no firm intention to receive a booster dose) limit the generalizability of the results presented to the Quebec population. Also, we must mention that the participants in the booster groups were relatively young and healthy. In Quebec, current recommendations concerning booster doses are aimed at people at risk, notably those aged 60 and over or with underlying health conditions. Although these recommendations were not in effect at the time of the focus groups, it would still have been interesting and informative to recruit people with these profiles. Nevertheless, our findings align with data collected from web surveys among large samples conducted during the same period in the province [39].

In conclusion, future efforts to enhance COVID-19 vaccine acceptance and uptake should focus on building confidence. As new COVID-19 vaccination recommendations and vaccine formulas emerge, public health authorities should recognize that the acceptability of COVID-19 vaccination tends to decrease with each additional dose. The COVID-19 "vaccine fatigue" presents a real challenge for public health vaccine promotion efforts. Public health authorities' communication efforts should focus on providing relevant information and addressing vaccine-specific concerns to maintain trust in vaccination programs.

## Supporting information

**S1 Appendix. Focus groups interview guides.**
(DOCX)

## Acknowledgments

The authors thank all the participants in the focus groups who openly shared their perceptions and experiences. We also want to thank Angèle Larivière for her work in reviewing transcriptions from Léger.

## Author Contributions

**Conceptualization:** Eve Dubé.

**Formal analysis:** Catherine Pelletier.

**Funding acquisition:** Eve Dubé.

**Project administration:** Dominique Gagnon.

**Supervision:** Eve Dubé.

**Validation:** Dominique Gagnon.

**Writing – original draft:** Catherine Pelletier, Dominique Gagnon.

**Writing – review & editing:** Eve Dubé.

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
