## [Decision Letter · Decision Letter 0]

4 Sep 2023

PONE-D-23-23031“It's not that I don't trust vaccines, I just don't think I need them”: perspectives on COVID-19 vaccinationPLOS ONE

Dear Dr. Dubé,

Thank you for submitting your manuscript to PLOS ONE. After careful consideration, we feel that it has merit but does not fully meet PLOS ONE’s publication criteria as it currently stands. Therefore, we invite you to submit a revised version of the manuscript that addresses the points raised during the review process.

We look forward to receiving your revised manuscript.

Kind regards,

Ouoba Kampadilemba, Ph.D.

Academic Editor

PLOS ONE

Journal Requirements:

Additional Editor Comments:

Just after the acknowledgements section, add and complete the following statements: "Funding"; "Conflict of interest"; "Ethics approval"; "Consent to participate"; "Data availability"; "Author contributions".

Reviewers' comments:

Reviewer's Responses to Questions

**Comments to the Author**

1. Is the manuscript technically sound, and do the data support the conclusions?

Reviewer #1: Yes

Reviewer #2: Yes

2. Has the statistical analysis been performed appropriately and rigorously? 

Reviewer #1: N/A

Reviewer #2: N/A

3. Have the authors made all data underlying the findings in their manuscript fully available?

Reviewer #1: No

Reviewer #2: No

4. Is the manuscript presented in an intelligible fashion and written in standard English?

Reviewer #1: Yes

Reviewer #2: Yes

5. Review Comments to the Author

Reviewer #1: Thank you for the opportunity to review your paper. I have suggested some revisions, with detailed comments below.

Abstract

- please define the term “primo-vaccination”

Introduction

- Good to give a brief description of Quebec (e.g. second largest province in…)

Methods

- Line 126 – I’m not familiar with the term “professional animator” – please define

- If you haven’t already, please provide your semi-structured interview guide in the Supplementary materials

- Include how you assessed intention in your participants (i.e. how did you collect the data shown in Table 1)

- Your analysis seems to be thematic analysis rather than content analysis. Please describe your analysis in more detail in the methods, providing references where relevant, to help the reader better understand the method you used.

Results

- In your inclusion criteria, you state that parents must “3) not have a firm intention to have them vaccinated at this time.” (Methods, line 119). But in Table 1, 4 of your participants are listed as having positive intention to vaccinate their child. This seems to contradict your inclusion criteria – please explain.

- Please explain why you had 2 participants with unspecified intention in your booster dose groups (see Table 1), especially when your inclusion criteria states that participants must “2) not to intend or be uncertain about receiving 122 an additional dose in the fall of 2022.” (Methods, line 122) How did you include these 2 participants if you did not know their intention?

- at the top of the results, it would be helpful to the reader to describe how many themes you found

- Line 161 – please rephrase, this sounds like women had been pregnant since Dec 2020 – impossible! “COVID-19 vaccine safety was also an important source of concern, especially among female participants who had been [BECOME?] pregnant since the launch of the COVID-19 vaccination campaign.”

- Line 181 – “Parents who reported an intention to have their child(ren) vaccinated against COVID-19…” - see previous comment, it’s not clear why you included these participants who did not meet your inclusion criteria

- Line 190 – same as above comment, it’s not clear why you included these participants who did not meet your inclusion criteria

- Box 1 – it would help the reader to put the booster quotes together

- Line 211 – is this section just about routine childhood vaccinations? Please be clearer

- Line 244 “However, other participants emphasized that the level of trust held no significance in the implementation of certain measures requiring compliance or adherence. For these participants, trust played a minimal role in determining the level of adherence.” Can you give an example here? It’s not clear to me what you are talking about here.

Discussion

- Line 309 “Both the acceptance of COVID-19 vaccination for young children and the willingness to receive a booster dose were found to be low…” – this statement is problematic, you did not aim to measure acceptance in this study but rather to understand the reasons behind low acceptance. Please consider rephrasing.

- It’s not necessary to repeat your results at the beginning of the Discussion section to this extent

- Rather than results, it would be more useful to see further comparisons with the existing literature in the discussion – how did your results agree or disagree with previously published works?

- Line 332 “Complacency… was also a major determinant of low COVID-19 vaccine acceptance among parents in this study” – I’m not sure ‘complacency’ is the correct term here. Yes the parents perceived the disease risk as low and did not see vaccination as necessary, but they were responding to the data, which showed that the disease risk WAS low for the 6 month – 4 year age group. Complacency implies a lack of awareness, overlooking problems and having a false sense of security.

- One potential limitation of your study that you do not mention is that you only included 1 person 55 years or older – as older people are the main beneficiaries of booster doses, it would have been informative to understand these perspectives

Reviewer #2: This is an interesting and timely paper as many countries are confronted with low demand for Covid-19 vaccines especially with a low perceived risk as the pandemic receded.

The methodology is well explained, the results well structured, and the discussion concise and expanded to include public health implications.

A few comments:

1) On lines 116-123, authors list eligibility criteria to be part of FGDs. However, there are no details on how/when these criteria were applied and verified. Was that at the recruitment stage? Was that part of the online survey that was sent to identify potential participants? Please explain further.

2) On lines 167-69, authors alluded to apprehensions about the lack of transparency from private vaccine manufacturers about vaccine safety and efficacy. But in the body of the results including illustrative quotes, there is no futher information or detail about this fact. Can you include a quote which says that?

3) Line 251: the term lockdown has been used, but elsewhere in the document you use the term confinement. Are they equivalent? If so, harmonize.

4) Line 286: "Although the abundance of information disseminated during the vaccination campaign, most participants in both the 6-months-4-years and booster dose groups felt that...". Shouldn't although be replaced by despite for more clarity in the sentence?

5) Line 343-4: As vaccine confidence proves to be volatile and subject to fluctuation over time.....Even if your findings also point to that, you can/should insert a reference to back that statement. For instance, consider inserting after "over time" this recent study in Europe: Sabat I et al; Vaccine hesitancy comes in waves: Longitudinal evidence on willingness to vaccinate against COVID-19 from seven European countries. Vaccine. 2023 Aug 14;41(36):5304-5312

6) Finally, the title of the paper is based on a quote ("It's not that I don't trust vaccines, I just don't think I need them") which reflects an important theme that emerged form your investigations. While I support the title as is, I was expecting as a reader to come across an illustrative quote in the body of the results that include the excerpt used in the titlled, but there was no such quote. I suggest you add the result that inspired the title.

6. PLOS authors have the option to publish the peer review history of their article (what does this mean?). If published, this will include your full peer review and any attached files.

Reviewer #1: No

Reviewer #2: **Yes: **Dr Lassane Kabore

---

## [Author Response · Author response to Decision Letter 0]

4 Oct 2023

POINT-BY-POINT RESPONSES TO COMMENTS FROM REVIEWERS

Lines are provided from the clean version of the manuscript.

Journal requirements and editor comments

 • Submitted documents have been revised to meet PLOS ONE criteria for text and title page format.

2. In your Data Availability statement, you have not specified where the minimal data set underlying the results described in your manuscript can be found. 

 • We understand that to ensure the transparency and reproducibility of research, a minimal data set must be made available. However, due to requirements by our Research ethics review board, we cannot make audio recordings or transcripts available. We have modified our Data Availability statement to better explain these issues. We have, however, added the interview guide as a Supporting Information file.

3. Please review your reference list to ensure that it is complete and correct. 

 • The reference list has been reviewed and updated.

4. Just after the acknowledgments section, add and complete the following statements: "Funding"; "Conflict of interest"; "Ethics approval"; "Consent to participate"; "Data availability"; "Author contributions".

 • These sections have been added to the manuscript.

Reviewer #1

Thank you for the opportunity to review your paper. I have suggested some revisions, with detailed comments below.

 • We thank the reviewer for the careful review of our manuscript. It helped improve our manuscript.

1. Abstract: please define the term “primo-vaccination”

 • In order to avoid making the abstract too long with a definition, we have replaced “primo-vaccination” with "first and second doses".

 o Page 1, line 31: While the first and second doses were well-accepted among adults and vaccine uptake was above 90%...

2. Introduction: Good to give a brief description of Quebec (e.g. second largest province in…)

 • Thanks for the relevant comment. We've added a few lines about Quebec and the way healthcare is managed in Canada to further introduce the context of the study.

 o Page 2, lines 63-67: Quebec is the second most populous province in Canada and is predominantly French-speaking. Canada is a federation and healthcare is under provincial and territorial jurisdiction. While national authorities were tasked with setting public health guidelines and overseeing vaccine supply and distribution during the pandemic, each province issued its own vaccination recommendations and managed its vaccination campaign. In Quebec,…

3. Methods: Line 126 – I’m not familiar with the term “professional animator” – please define.

 • The moderation was entrusted to a professional specialized in facilitating group discussions. This professional came from the same research firm responsible for recruiting the participants. We have made these clarifications in the methodology.

 o Pages 5-6, lines 135-137: Facilitation was handled by a professional with expertise in moderating focus groups, hailing from the same specialized research firm that managed the recruitment process.

4. Methods: If you haven’t already, please provide your semi-structured interview guide in the Supplementary materials

 • The interview guide is attached to this revised version (separate file labeled S1 Appendix).

5. Methods: Include how you assessed intention in your participants (i.e. how did you collect the data shown in Table 1)

 • We thank the reviewer for his comment. We had briefly listed the topics covered in the discussions, but we understand that it wasn't obvious that participants were asked about their intentions during the focus groups. We have added a sentence to clarify this.

 o Pages 6, lines 138-142: A semi-structured interview guide covering topics related to trust in government and public health authorities during the pandemic, vaccination (COVID-19 and routine), and information available on COVID-19 vaccination was used. Participants were invited to share whether they intended or not to vaccinate their child(ren) aged 5 and under or to receive a booster dose themselves.

6. Methods: Your analysis seems to be thematic analysis rather than content analysis. Please describe your analysis in more detail in the methods, providing references where relevant, to help the reader better understand the method you used.

 • Thank you to the reviewer for this comment. We have indeed carried out a thematic content analysis (TCA). We have added a few details, as well as a reference, to further illustrate our methodological approach. 

 o Page 6, lines 146-150: A thematic content analysis was conducted using NVivo 12 software to identify and interpret themes within the dataset [16]. Inductive and deductive approaches were used to generate codes. While certain themes were pre-identified based on the discussion guide, additional themes emerged during the analysis to capture participants’ perspectives. 

7. Results: In your inclusion criteria, you state that parents must “3) not have a firm intention to have them vaccinated at this time.” (Methods, line 119). But in Table 1, 4 of your participants are listed as having positive intention to vaccinate their child. This seems to contradict your inclusion criteria – please explain.

 • We thank the reviewer for his comment. We acknowledge that the expression "not have a firm intention" may be unclear. To be included in the study, parents could either not intend to have their children vaccinated, be hesitant, or unsure if they were going to have their children vaccinated. We have revised our methodology to explicitly state this, and we believe that the data presented in the table now makes more sense.

 o Page 5, lines 124-129: Thus, to be eligible to participate in the focus groups on the acceptability of COVID-19 vaccination for children, participants needed to meet the following criteria at the time of recruitment: 1) they had to be parents with at least one child under the age of 5, 2) their child(ren) must not have received the COVID-19 vaccination, and 3) they had to either express no intention, be hesitant, or be unsure about vaccinating their child(ren). 

8. Results: Please explain why you had 2 participants with unspecified intention in your booster dose groups (see Table 1), especially when your inclusion criteria states that participants must “2) not to intend or be uncertain about receiving an additional dose in the fall of 2022.” (Methods, line 122) How did you include these 2 participants if you did not know their intention?

 • The inclusion criteria for participants in the booster dose focus groups were assessed at the time of recruitment. Participants had to not intend, be hesitant, or unsure about receiving a booster dose to participate in the focus group. As mentioned in our response to comment #5, intention was also questioned during the discussion. In the manuscript, we report intention as stated by the participants during the focus groups (Table 1). It is therefore possible that participants expressed a different opinion between the recruitment and data collection. We understand that this can be confusing and have added an explanatory footnote to Table 1.

 o Page 5, lines 129-133: For the focus groups on the acceptability of a booster dose of a COVID-19 vaccine, participants had to meet these criteria at the time of recruitment: 1) they needed to have received either two or three doses of a COVID-19 vaccine, and 2) they had to either express no intention, be hesitant, or be unsure about receiving an additional dose in the fall of 2022.

 o Page 8, line 170: a As reported during the focus groups.

9. Results: at the top of the results, it would be helpful to the reader to describe how many themes you found

 • The themes refer mainly to the topics covered in the discussions. For the convenience of readers, we added additional information at the beginning of the results.

 o Page 8, lines 171-178: The main topics covered in the focus groups were aligned with the themes of the semi-structured interview guide. The findings are presented in an aggregated manner and are divided into four sections: opinion (including intention) towards COVID-19 vaccines, perceptions of routine vaccines since the start of the pandemic, trust in authorities and perceptions of vaccine campaign and related measures, and attitudes and preferences in terms of information about COVID-19 vaccines. We paid specific attention to similarities and differences in discourses about COVID-19 booster doses and the vaccination of children against COVID-19.

10. Results: Line 161 – please rephrase, this sounds like women had been pregnant since Dec 2020 – impossible! “COVID-19 vaccine safety was also an important source of concern, especially among female participants who had been [BECOME?] pregnant since the launch of the COVID-19 vaccination campaign.”

 • Thanks to the reviewer for identifying this typo. We have made the correction.

 o Page 8, line 187: COVID-19 vaccine safety was also an important source of concern, especially among participants who had become pregnant after the launch of the COVID-19 vaccination campaign.

11. Results: Line 181 – “Parents who reported an intention to have their child(ren) vaccinated against COVID-19…” - see previous comment, it’s not clear why you included these participants who did not meet your inclusion criteria

 • We hope that our response to comment #7 and the changes made in the manuscript will answer this question.

12. Results: Line 190 – same as above comment, it’s not clear why you included these participants who did not meet your inclusion criteria

 • We hope that our response to comment #7 and the changes made in the manuscript will answer this question.

13. Results: Box 1 – it would help the reader to put the booster quotes together

 • The quotes were presented in the order of the topics covered in each results section. We understand, however, that this may not be obvious to the reader. Since we want to highlight similarities or differences between the groups (parents vs. booster), we have therefore chosen to do as suggested by the reviewer and place the quotes from the booster groups together, as well as those from the parent groups. We have, however, retained a single box per section, since we specify the participant and the group to which each quote belongs. To be consistent throughout the manuscript, we have placed the excerpts from the parent groups first, followed by those from the booster groups.

 o Box 1, page 11

 o Box 2, page 13

 o Box 3, pages 16-17

14. Results: Line 211 – is this section just about routine childhood vaccinations? Please be clearer

 • We acknowledge that this section could create some confusion for the reader as most routine vaccines are recommended in infants and children. The question was asked to all participants, including in COVID-19 booster dose discussions, and was not asking specifically about routine childhood vaccines. As many participants in the COVID-19 booster dose were also parents of young children, they naturally talked about their experience with childhood vaccines. Most participants were not part of the groups for which other routine vaccines are recommended in the Quebec immunization program (i.e., flu recommended to 75 years and older or people with chronic medical conditions, pneumococcal vaccine recommended to 65 years and older, shingles recommended to people aged 80 years). However, some participants talked about influenza vaccines in adults. We have clarified the text and added a quote in box 2 to further illustrate this point.

 o Page 12, lines 237-239: Despite concerns about COVID-19 vaccines and low intentions, opinions of other routine vaccines remained the same as before the pandemic for all participants interviewed, including those involved in COVID-19 booster dose discussions.

 o Page 12, lines 242-245: Despite this high level of confidence, some participants still had concerns about specific vaccines administered during childhood or adulthood (e.g., MMR vaccine because of its supposed link with autism in children, varicella vaccine because of its more recent availability, influenza vaccine because of its uneven efficacy). 

 o Page 12, lines 246-248: During the discussions, all parents from all four focus groups spontaneously affirmed that their child(ren) had received all the recommended vaccines up to now and that their vaccination records were up to date.

 o Box 2, page 13: But I don't take the flu shot because I'm a bit worried about it. […] Of course, because of my health condition… and it's clear that getting an injection... It's like injecting an entity into your body, and I think you have to be well-informed. (Participant 5, COVID-19 booster dose, Focus group 1)

15. Results: Line 244 “However, other participants emphasized that the level of trust held no significance in the implementation of certain measures requiring compliance or adherence. For these participants, trust played a minimal role in determining the level of adherence.” Can you give an example here? It’s not clear to me what you are talking about here.

 • We thank the reviewer for this comment and have rephrased part of the sentence and added an example to make our idea clearer.

 o Page 14, lines 274-277: For these participants, trust played a minimal role in determining the level of adherence, as they were following the rules implemented as part of the state of emergency (e.g., curfew, travel bans), regardless of their opinion of the authorities.

16. Discussion: Line 309 “Both the acceptance of COVID-19 vaccination for young children and the willingness to receive a booster dose were found to be low…” – this statement is problematic, you did not aim to measure acceptance in this study but rather to understand the reasons behind low acceptance. Please consider rephrasing.

 • We acknowledge that this statement can be problematic. We have indeed explored the intention, so we’ve rephrased it accordingly. 

 o Page 20, lines 339-340: Both the intention to vaccinate children against COVID-19 and the willingness to receive a booster dose were found to be low, with common underlying issues.

17. Discussion: It’s not necessary to repeat your results at the beginning of the Discussion section to this extent

 • In response to comments #17 and 18, we've worked to enhance the depth of our discussion. We have addressed concerns regarding results redundancy by summarizing and incorporating additional comparisons with the literature. We believe we have strengthened the quality of the discussion by doing so. 

 o See pages 20-24 for the discussion

18. Discussion: Rather than results, it would be more useful to see further comparisons with the existing literature in the discussion – how did your results agree or disagree with previously published works?

 • See our response to the previous comment.

19. Discussion: Line 332 “Complacency… was also a major determinant of low COVID-19 vaccine acceptance among parents in this study” – I’m not sure ‘complacency’ is the correct term here. Yes the parents perceived the disease risk as low and did not see vaccination as necessary, but they were responding to the data, which showed that the disease risk WAS low for the 6 month – 4 year age group. Complacency implies a lack of awareness, overlooking problems and having a false sense of security.

 • We agree with the reviewer comment regarding the risk of COVID-19 for young children and healthy adults who already have received the first two doses. The word complacency was removed from this section of the discussion.

 • See also our response to #17.

20. Discussion: One potential limitation of your study that you do not mention is that you only included 1 person 55 years or older – as older people are the main beneficiaries of booster doses, it would have been informative to understand these perspectives

 • We thank the reviewer for this very relevant comment. This is indeed an important limitation, especially considering the current booster dose recommendations in Quebec, which target at-risk individuals (people aged 60 and over, people with underlying health conditions, etc). We have therefore added a few lines on this subject in the limitations section.

 o Page 23, lines 410-415: Also, we should mention that the participants in the booster groups were relatively young and healthy. In Quebec, current recommendations concerning booster doses are aimed at people at-risk, notably those aged 60 and over or with underlying health conditions. Although these recommendations were not in effect at the time of the focus groups, it would still have been interesting and informative to recruit people with these profiles.

Reviewer #2

This is an interesting and timely paper as many countries are confronted with low demand for Covid-19 vaccines especially with a low perceived risk as the pandemic receded. The methodology is well explained, the results well structured, and the discussion concise and expanded to include public health implications.

 • We thank the reviewer for his thoughtful comments that helped improve our manuscript.

1. On lines 116-123, authors list eligibility criteria to be part of FGDs. However, there are no details on how/when these criteria were applied and verified. Was that at the recruitment stage? Was that part of the online survey that was sent to identify potential participants? Please explain further.

 • We thank the reviewer for this comment. Participants were pooled and selected by the research firm. The inclusion criteria were applied and verified at the stage of recruitment, by the firm. 

 o Page 5, lines 122-124: The specialized research firm was responsible for contacting, selecting, and recruiting participants who met the inclusion criteria. Thus, to be eligible to participate in the focus groups…

2. On lines 167-69, authors alluded to apprehensions about the lack of transparency from private vaccine manufacturers about vaccine safety and efficacy. But in the body of the results including illustrative quotes, there is no further information or detail about this fact. Can you include a quote which says that?

 • We thank the reviewer for this observation. In writing the manuscript, we took care to report all the participants' comments, even those less frequently discussed. However, as this was noted by only a few participants, we feel that including a quote for each idea or statement give this comment too much weight as this was not a major point of the discussion. For this reason, we have removed the sentence.

3. Line 251: the term lockdown has been used, but elsewhere in the document you use the term confinement. Are they equivalent? If so, harmonize.

 • We have chosen the term "lockdown", which better reflects the situation during the pandemic. We have also chosen to replace "confinement" with "stay at home" in an excerpt to reflect the participant's comments.

 o Box 2, page 13: Both my children are up to date with their vaccinations. My youngest, in the beginning, during the big lockdown, the first one, that's when her vaccines were most due.

 o Box 3, page 17: Trust waned a bit. At the beginning, they said that with the vaccine, everything would be fine, we'd go back to the way things were before. Then you get the vaccine, and they say, "Oh no, you need a 2nd dose and stay at home".

4. Line 286: "Although the abundance of information disseminated during the vaccination campaign, most participants in both the 6-months-4-years and booster dose groups felt that...". Shouldn't although be replaced by despite for more clarity in the sentence?

 • Indeed. We made the correction according to the reviewer’s suggestion.

 o Page 18, line 316: Despite the abundance of information disseminated during the vaccination campaign, most participants in both the 6-months-4-years and booster dose groups felt that there was still a lack of truly useful information that would support informed decision-making (e.g., lack of clarity regarding the benefits of vaccinating under 5 years old and the effectiveness of one or more booster doses).

5. Line 343-4: As vaccine confidence proves to be volatile and subject to fluctuation over time.....Even if your findings also point to that, you can/should insert a reference to back that statement. For instance, consider inserting after "over time" this recent study in Europe: Sabat I et al; Vaccine hesitancy comes in waves: Longitudinal evidence on willingness to vaccinate against COVID-19 from seven European countries. Vaccine. 2023 Aug 14;41(36):5304-5312

 • We thank the reviewer for the reference. We believe it is relevant and have added it where suggested. We have also edited the reference list accordingly.

 o Page 21, line 382: As vaccine confidence proves to be volatile and subject to fluctuation over time [34], …

6. Finally, the title of the paper is based on a quote ("It's not that I don't trust vaccines, I just don't think I need them") which reflects an important theme that emerged form your investigations. While I support the title as is, I was expecting as a reader to come across an illustrative quote in the body of the results that include the excerpt used in the title, but there was no such quote. I suggest you add the result that inspired the title.

 • We thank the reviewer for this comment. We agree that it’s important to include the verbatim used for the article title in the results. We have chosen to replace an existing verbatim with this one (from the same group).

 o Box 1, page 11: I've had my 2 doses, but as far as booster doses go, I don't see the point. It's not that I don't trust vaccines, but I don't think I need them. (Participant 7, COVID-19 booster dose, Focus group 2)

---

## [Editor Report · Decision Letter 1]

18 Oct 2023

“It's not that I don't trust vaccines, I just don't think I need them”: perspectives on COVID-19 vaccination

PONE-D-23-23031R1

Dear Dr. Dubé

We’re pleased to inform you that your manuscript has been judged scientifically suitable for publication and will be formally accepted for publication once it meets all outstanding technical requirements.

Kind regards,

Ouoba Kampadilemba, Ph.D.

Academic Editor

PLOS ONE

---

## [Editor Report · Acceptance letter]

20 Oct 2023

PONE-D-23-23031R1 

*“It's not that I don't trust vaccines, I just don't think I need them”:* perspectives on COVID-19 vaccination 

Dear Dr. Dubé:

I'm pleased to inform you that your manuscript has been deemed suitable for publication in PLOS ONE. Congratulations! Your manuscript is now with our production department. 

Kind regards, 

on behalf of

Dr. Ouoba Kampadilemba 

Academic Editor

PLOS ONE